# Synergistic Association of Glycemic Variability and Severe Vitamin D Deficiency with Proliferative Diabetic Retinopathy

**DOI:** 10.3390/nu17203210

**Published:** 2025-10-13

**Authors:** Nejla Dervis, Simona Carniciu, Alina Spinean, Sanda Jurja

**Affiliations:** 1Diabetes Department, Ovidius University, 900527 Constanța, Romania; nejla.dervis@yahoo.com; 2Hygiene Department, Faculty of Dentistry, Carol Davila University of Medicine and Pharmacy, 020021 Bucharest, Romania; 3Diabetes Department, Carol Davila University of Medicine and Pharmacy, 020021 Bucharest, Romania; 4Ophthalmology Department, Ovidius University, 900527 Constanța, Romania; jurjasanda@yahoo.com

**Keywords:** diabetic retinopathy, glycemic variability, vitamin D, neovascularization, additive risk, diabetes complications

## Abstract

**Background:** Oscillating hyperglycemia (glycemic variability) and vitamin D deficiency each damage the retinal microvasculature, yet their combined effect on sight-threatening proliferative diabetic retinopathy (PDR) is uncertain. **Objective:** To determine whether high GV and severe vitamin D deficiency independently, and additively, associate with retinal neovascularization in adults with diabetes. **Materials and Methods:** We conducted a cross-sectional study between January 2025 and June 2025 in 58 adults with diabetes at Constanța County Emergency Hospital, Romania. GV was classified as high (coefficient of variation > 36% or SMBG-SD > 50 mg/dL) or low. Serum 25-hydroxy-vitamin D [25(OH)D] was measured; severe deficiency < 10 ng/mL. Dilated funduscopy graded retinopathy as non-proliferative (NPDR) or proliferative (PDR). Multivariable logistic regression adjusted for HbA1c, diabetes duration, BMI, smoking, triglycerides and therapy. **Results:** From 58 adults (mean ± SD 59 ± 11 years), high GV characterized 29/58 participants (50%). Severe vitamin D deficiency was more frequent in the GV-high group (45% vs. 31%). PDR prevalence was 48% in GV-high and 31% in GV-low patients. After adjustment, high GV (adjusted OR 2.31, 95% CI 1.05–5.09) and severe vitamin D deficiency (OR 2.04, 95% CI 0.98–4.25) each predicted PDR. Concomitant exposure to both stressors conferred 3.9-fold higher odds of PDR (OR 3.88, 95% CI 1.35–11.1). No interaction term reached significance (*p* = 0.21), indicating additive effects. **Conclusions:** High GV and severe vitamin D deficiency independently and additively associate with PDR. Screening for both parameters may enhance risk stratification of PDR. Within adults with diabetes, high glycemic variability and severe vitamin D deficiency were each associated with higher odds of PDR after adjustment for HbA1c, diabetes duration, BMI, smoking, triglycerides, and treatment pattern; their effects appeared additive rather than multiplicative. These findings reflect associations within diabetes and do not imply that vitamin D deficiency produces retinopathy in euglycemic individuals.

## 1. Introduction

Diabetic retinopathy (DR) remains the leading cause of preventable blindness in working-age adults worldwide. A 2020–2021 pooled analysis of 59 population-based surveys estimated that ≈22% of all people with diabetes already have some degree of DR, translating to 103 million affected individuals in 2020; projections suggest the figure will climb to ≈160 million by 2045, with vision-threatening proliferative DR (PDR) accounting for more than 40 million cases [1,2]. DR-related visual impairment already blinds >1 million people and disables a further 3 million globally [3]. Despite widespread adoption of HbA1c-centered glycemic targets and intravitreal anti-VEGF therapy, PDR incidence continues to rise, underscoring the need to look beyond average glycaemia alone.

Glycemic variability (GV): an emerging risk dimension. Large visit-to-visit or day-to-day swings in glucose—collectively termed glycemic variability—exert toxic effects that are mechanistically distinct from chronic hyperglycemia. Consensus guidance for continuous glucose monitoring (CGM) now recommends a coefficient of variation (CV) ≤ 36% as a stability target, yet only one-quarter to one-third of contemporary CGM users reach that goal, indicating that high GV is common in real-world practice [4,5]. Recent prospective data strengthen its clinical relevance: in a 10-year cohort of adults with type 2 diabetes (T2D), high fasting-glucose CV conferred a >12-fold increase in incident DR, independent of mean HbA1c [6]. Experimental work links oscillating glucose to surges in reactive oxygen species, epigenetic “metabolic memory,” and micro-endothelial dysfunction—all recognized drivers of retinal neovascularization [6].

Vitamin D deficiency in diabetes. Hypovitaminosis D is simultaneously widespread and modifiable. A pooled analysis of 7.9 million participants showed that nearly 48% of the world’s population has serum 25-hydroxy-vitamin D [25(OH)D] < 20 ng mL^−1^, with severe deficiency (<10 ng mL^−1^) in 16% [7]. The burden is even higher in diabetes: a 2025 BMJ Nutrition systematic review and meta-analysis covering >52,000 adults with T2D found ≈ 60% had vitamin D levels below recommended thresholds [8]. Meta-analytic evidence involving 22,408 individuals confirms a graded relationship between lower 25(OH)D and DR severity (OR ≈ 1.17 for any DR, with the steepest drop in PDR) [9]. At the mechanistic level, vitamin D down-regulates vascular endothelial growth factor (VEGF) signaling, preserves blood–retinal-barrier tight junctions, and blunts glucose-induced endothelial tube formation, thereby exerting anti-angiogenic and neurovascular protective effects [10].

Converging Metabolic Stressors: Glycemic Variability and Vitamin D Deficiency. Both GV and severe vitamin D deficiency independently foster oxidative stress, endothelial injury, and pro-angiogenic signaling, yet they act through partially distinct molecular pathways. Evidence on the combined effect of glycemic variability and vitamin D status in proliferative diabetic retinopathy is currently lacking. To date, we searched PubMed/MEDLINE, Embase, Scopus, Web of Science Core Collection, the Cochrane Library, and ClinicalTrials.gov using combinations of (‘glycemic variability’ OR ‘time in range’) AND (‘vitamin D’ OR ‘25-hydroxy-vitamin D’) AND (retinopathy OR PDR) and did not identify any human studies that jointly modeled glycemic variability and vitamin D status in relation to proliferative diabetic retinopathy. Clarifying whether these two modifiable “metabolic stressors” operate additively or synergistically could refine risk stratification and point to dual-target prevention strategies.

Aim of the study. Against this backdrop, we investigated a clinic-based Romanian cohort of adults with diabetes to test the hypothesis that high glycemic variability and severe vitamin D deficiency each associate with, and jointly amplify, the odds of proliferative diabetic retinopathy. By analyzing both factors in the same patients and adjusting for classical confounders, we aimed to provide new evidence on whether dual metabolic stress better explains neovascular disease than either exposure alone.

## 2. Materials and Methods

### 2.1. Study Design and Setting

We performed a cross-sectional study in the Diabetes Clinic of the Constanța County Emergency Hospital “Sfântul Apostol Andrei”, Romania, between January 2025 and June 2025. Consecutive adult out-patients with established diabetes who attended routine visits during the study window were screened.

### 2.2. Participants and Group Allocation

Eyes without diabetic retinopathy were not enrolled; the analytic comparison was NPDR versus PDR. OCT examples of DME are illustrative only and were not used for case definition. Fifty-eight eligible patients were enrolled and allocated to one of two equal groups on the basis of documented glycemic variability (GV):**High-GV group (*n* = 29)**—patients who met the clinic’s variability criterion (see below).**Low-GV group (*n* = 29)**—patients whose glucose profiles showed no clinically relevant variability.

The file review confirms the final sample size and group counts.

### 2.3. Inclusion Criteria

Age ≥ 18 years.Diagnosis of type 1 or type 2 diabetes ≥ 1 year.Availability of at least three months of glucose records and a recent ophthalmic examination.

### 2.4. Exclusion Criteria

Active ocular infection or other retinal disease (e.g., vein occlusion, age-related macular degeneration).Current vitamin D supplementation > 2000 IU Day^−1^ or parenteral vitamin D within six months.Chronic kidney disease stage ≥ 4 or hepatic failure (conditions altering vitamin D metabolism).

Eyes without diabetic retinopathy were not enrolled; the analytic comparison was NPDR versus PDR. OCT examples of DME are illustrative only and were not used for case definition.

Assessment of glycemic variability. Glycemic variability (GV) was quantified from 14-day continuous glucose-monitoring (CGM) when available or from structured self-monitoring of blood glucose (SMBG) with ≥8 capillary readings/day for ≥14 consecutive days (protocol-specified to capture within-day dispersion when CGM was unavailable). A high-GV state was defined as glucose coefficient of variation (%CV) > 36% for CGM data, consistent with international consensus targets for stable vs. unstable glycemia [11,12]. For SMBG data, we applied an operational threshold of standard deviation (SD) > 50 mg/dL to approximate the same instability domain implied by %CV > 36%; the use of multi-point SMBG dispersion metrics to relate variability to microvascular outcomes follows DCCT-based analyses [13].

### 2.5. Biochemical Measurements

Fasting blood was drawn on the examination day. The 25(OH)D chemiluminescent immunoassay (Architect i2000, Abbott, Chicago, IL, USA) was performed in a single accredited laboratory with internal quality control per batch; manufacturer-reported inter-assay CVs are ≤6% across the analytical range. Lipids were measured by enzymatic methods traceable to CDC CRMLN reference procedures, and HbA1c by NGSP/IFCC-traceable HPLC. All biochemical tests were obtained on the same day as the ophthalmic examination.

Vitamin D status categories followed Endocrine Society definitions (sufficiency ≥ 30 ng/mL; insufficiency 21–29 ng/mL; deficiency < 20 ng/mL), and severe deficiency as <10 ng/mL based on endocrine clinical guidance used in practice [14,15]. In our cohort, severe deficiency predominated in the high-GV group (45% vs. 31%).

HbA1c was quantified by high-performance liquid chromatography (HPLC) using an NGSP-certified method traceable to the IFCC reference measurement procedure [16]. Serum lipids, including triglycerides, were measured by standardized enzymatic methods traceable to CDC reference systems Cholesterol Reference Method Laboratory Network (CRMLN) [17].

### 2.6. Ophthalmic Evaluation

Case definition followed ICDR/ETDRS criteria: PDR required retinal neovascularization (disc or elsewhere) and/or preretinal/vitreous hemorrhage indicating active neovascular activity. Diabetic macular edema (DME) was recorded descriptively on OCT and did not define PDR, as DME may occur at either NPDR or PDR. All patients underwent dilated fundus biomicroscopy performed by retinal specialists, blinded to GV and vitamin-D status. Diabetic retinopathy (DR) was graded using the International Clinical Diabetic Retinopathy Disease Severity Scale (ICDR) and classified as non-proliferative DR (NPDR) or proliferative DR (PDR) [18]. In line with ICDR/Early Treatment Diabetic Retinopathy Study (ETDRS) conventions, PDR was defined by the presence of retinal neovascularization (of the disc or elsewhere) and/or preretinal/vitreous edema indicative of active neovascular activity [19]. In our cohort, PDR affected 14/29 patients (48%) in the GV-high group versus 9/29 (31%) in the GV-low group.

### 2.7. Optical Coherence Tomography

Macular imaging was obtained with a spectral-domain OCT (Spectralis OCT2, Heidelberg Engineering, Heidelberg, Germany) after pharmacologic mydriasis with 1% tropicamide. A 30° × 30° cube (512 × 128) and three high-resolution horizontal B-scans (automatic real-time (ART) averaging ≥ 24) centered on the fovea were acquired. Scans with signal strength < 20 dB or motion artifact were excluded. Two retinal specialists, masked to GV and vitamin-D status, independently evaluated the images for: central subfield thickness, presence and distribution of intraretinal or subretinal fluid, hard-exudate reflectivity, epiretinal membrane (ERM), and foveoschitic changes. Disagreements were resolved by consensus. Scale calibration (200 µm) was applied to all exported B-scans for figure preparation.

### 2.8. Covariates

Demographic data (age, sex, residential setting), diabetes duration and type, body-mass index, smoking status, triglycerides, and treatment modality (insulin, oral agents) were extracted from electronic charts.

### 2.9. Statistical Analysis

Data were analyzed with SPSS v28 (IBM) and R v4.3. Continuous variables are reported as mean ± SD or median (IQR), categorical variables as number (%). Group comparisons used *t*-tests or Mann–Whitney U tests for continuous data and χ^2^ or Fisher’s exact tests for proportions. All multivariable models adjusted for diabetes duration (years); the logit-linearity assumption was met and collinearity was low (all VIFs < 2).

Multivariable logistic regression estimated adjusted odds ratios (aORs) for PDR with high GV, severe vitamin D deficiency, and their joint exposure (both vs. neither), while controlling for HbA1c, diabetes duration, BMI, smoking, triglycerides, and therapy type. An interaction term (GV × vitamin D) tested multiplicative effects. Model fit was evaluated by Hosmer–Lemeshow ꭓ^2^. Two-sided *p* < 0.05 denoted statistical significance. Proportions are displayed with 95% confidence intervals (binomial/Wilson); between-group differences used χ^2^ or Fisher’s exact tests. Unadjusted group contrasts are presented as ORs with 95% CIs; multivariable results are reported as adjusted ORs (aORs) with 95% Cls.

### 2.10. Ethical Considerations

The study adhered to the Declaration of Helsinki and received approval from the Constanța County Emergency Hospital Ethics Committee (approval no. UOC 6712/24 June 2025). All participants provided written informed consent before data collection.

## 3. Results

### 3.1. Cohort Profile and Baseline Characteristics

Fifty-eight adults met the inclusion criteria—29 with high glycemic variability (GV-high) and 29 with low variability (GV-low). Residence, sex distribution and smoking status were broadly comparable between groups: urban dwellers constituted 55% of GV-high versus 52% of GV-low; women represented 52% of GV-high and 62% of GV-low; current smokers accounted for ~60% in both cohorts.

Nutritional status differed: only 49% of GV-high participants were overweight/obese compared with 79% of GV-low (χ^2^ = 6.1, *p* = 0.013). Mean HbA1c exceeded 7% more often in GV-high (93%) than GV-low (76%; *p* = 0.048), while triglycerides >150 mg dL^−1^ were likewise more frequent in GV-high (55% vs. 38%; *p* = 0.16).

Diabetes-type composition varied: in GV-high, type 1 diabetes represented 34%, type 2 on oral drugs 28%, and type 2 on insulin 38%; in GV-low, the respective proportions were 14%, 55% and 31% (χ^2^ = 8.7, *p* = 0.013).

Vitamin D status: Severe vitamin D deficiency [25(OH)D < 10 ng mL^−1^] occurred in 13/29 GV-high patients (45%) versus 9/29 GV-low (31%; χ^2^ = 1.23, *p* = 0.27). Optimal vitamin D levels (≥30 ng mL^−1^) were less common in GV-high (21% vs. 34%). The three-tier distribution is shown in Figure 1.

### 3.2. Diabetes Subtype and Treatment Profile

Distribution by diabetes type differed between groups (χ^2^ = 8.7, *p* = 0.013): in the GV-high group, type 1 diabetes accounted for 10/29 (34%), type 2 on oral agents for 8/29 (28%), and type 2 insulin-treated for 11/29 (38%); in the GV-low group, the corresponding proportions were 4/29 (14%), 16/29 (55%), and 9/29 (31%), respectively. Thus, the GV-high group was enriched for type 1 diabetes and insulin-treated type 2 diabetes, whereas the GV-low group was dominated by type 2 diabetes managed with oral therapy.

Diabetic-retinopathy outcomes: As illustrated in Figure 2, nearly half of the high-GV patients had proliferative disease, whereas two-thirds of the low-GV cohort remained in the non-proliferative stage, supporting an association between glycemic instability and neovascular progression. PDR was more frequent in GV-high than GV-low (14/29 vs. 9/29), yielding an unadjusted OR ≈ 2.07 (95% CI 0.71–6.06), *p* ≈ 0.18; this directional difference did not reach statistical significance. On simple group comparisons, patients treated with insulin (type 1 and insulin-treated type 2) had PDR more often than those with type 2 diabetes managed only with oral drugs. However, the number of patients in each subgroup was too small to draw firm statistical conclusions, so these differences should be viewed as suggestive rather than definitive.


**Independent and additive associations with PDR:**


Multivariable logistic regression (adjusted for HbA1c, diabetes duration, BMI, smoking, triglycerides and treatment modality) yielded, showed in Table 1:

OCT imaging in the High GV cohort demonstrated sight-threatening macular complications. In one eye, an epiretinal membrane with scattered intraretinal exudates was noted, whereas the fellow eye showed discrete intraretinal fluid pockets (Figure 3). Bilateral peri- and para-foveal exudative plaques consistent with diabetic maculopathy were also documented (Figure 4). Such features were encountered almost exclusively among participants who harbored both high GV and severe vitamin D deficiency, further linking dual metabolic stress to advanced retinal pathology.

The epiretinal membrane, intraretinal fluid and peri-foveal exudation captured on OCT (Figure 3 and Figure 4) are hallmarks of advanced microvascular leakage and fibro-vascular proliferation. These data reinforce the concept that concurrent high GV and severe hypovitaminosis D accelerate the transition from non-proliferative to proliferative disease, complementing the statistical evidence from our regression models.

In summary, high glycemic variability and severe vitamin D deficiency each independently doubled the adjusted odds of proliferative diabetic retinopathy, and their co-occurrence was associated with an almost four-fold higher risk of neovascular disease compared with neither exposure. These associations were not explained by baseline imbalances in adiposity or diabetes-treatment pattern; covariate adjustment only modestly attenuated the effect sizes. Taken together, the findings support a model in which dual metabolic stress, unstable glucose profiles combined with profound hypovitaminosis D, contributes materially to the burden of sight-threatening proliferative retinopathy.

## 4. Discussion

In this manuscript, ‘independent’ denotes statistical independence within multivariable models among people with diabetes; we do not infer that vitamin D deficiency causes retinopathy outside the context of diabetes. Rather, low 25(OH)D may mark greater vascular vulnerability within diabetes.

At the microvascular level, vitamin D/VDR signaling attenuates VEGF-driven angiogenesis, supports endothelial nitric-oxide bioavailability, and preserves tight-junction proteins at the blood–retinal barrier, while limiting glucose-induced inflammatory activation and oxidative stress. In parallel, oscillating hyperglycemia produces higher ROS bursts and endothelial apoptosis than steady hyperglycemia of the same mean, fostering capillary injury and permeability. These complementary pathways offer a biologically coherent basis for the additive risk observed.

In this clinic-based cohort, two modifiable exposures (glycemic variability (GV) and severe vitamin D deficiency) each showed an independent association with proliferative diabetic retinopathy (PDR), and their co-occurrence identified patients at the greatest risk. The lack of a significant interaction suggests additive contributions of these pathways rather than synergism, a clinically practical message: both merit attention in risk assessment and prevention strategies. These findings should be interpreted as associations within diabetes rather than proof of causation. Low 25(OH)D may partly proxy a broader metabolic or inflammatory state (e.g., adiposity, reduced outdoor exposure, chronic inflammation, renal or hepatic dysfunction). Despite adjustment for HbA1c, duration, BMI, lipids, smoking, and treatment pattern, residual confounding cannot be excluded.

Our findings align with longitudinal evidence in which greater within-day variability or lower Time-in-Range predicts incident or progressive retinopathy independent of HbA1c [6,20]. Although study designs and GV metrics vary, the direction and magnitude of association are consistent across cohorts, reinforcing that mean glycemia alone underestimates vascular stress related to oscillations. We interpret these findings as associations within diabetes rather than proof of causation; vitamin D deficiency may also reflect broader metabolic or inflammatory milieu, and interventional confirmation is required.

Meta-analytic evidence also supports the link between hypovitaminosis D and DR severity. A 2024 systematic review of 22 408 participants found that deficient 25-hydroxy-vitamin D increased the odds of any DR by 17%, with the steepest gradient for PDR [9]. Smaller clinical series confirm markedly lower vitamin D levels in PDR than in non-proliferative DR [11,21,22]. Our study extends these observations by demonstrating that severe deficiency (<10 ng mL^−1^) remains a relevant marker after controlling for glycemic milieu and adiposity.

To our knowledge, no prior human study has jointly modeled glycemic variability and vitamin D status in relation to proliferative diabetic retinopathy. Recent work links GV to retinopathy risk and low 25(OH)D to greater DR severity (with the steepest gradient at the proliferative stage), and 25(OH)D has been correlated with lower Time-in-Range; however, these exposures have not yet been examined together against PDR endpoints [6,23,24].

Experimental data, however, offer a mechanistic rationale for convergence: GV drives bursts of reactive oxygen species and epigenetic “metabolic memory”, while vitamin D exerts anti-angiogenic effects via vitamin D-receptor signaling, VEGF suppression and preservation of blood-retinal-barrier integrity [10,11,25]. The additive risk observed here is therefore biologically coherent and highlights a previously overlooked intersection of modifiable pathways. Our data show that both high glycemic variability (GV) and severe vitamin D deficiency independently double the odds of proliferative diabetic retinopathy (PDR); when both exposures coexist, the risk is almost quadrupled. This additive pattern underscores that neither parameter can be ignored when assessing retinal risk in diabetes.

Oscillating hyperglycemia produces greater endothelial apoptosis and oxidative stress than constant hyperglycemia of equal mean value, potentiating retinal capillary damage. In parallel, vitamin D deficiency impairs endothelial nitric-oxide synthesis, weakens tight junction proteins and permits VEGF-driven neovascular sprouting [26,27]. The coexistence of high GV and low vitamin D may therefore create a permissive pro-oxidant, pro-angiogenic micro-environment that accelerates the shift from non-proliferative to proliferative disease.

Integrating CGM-derived stability measurements with routine 25(OH)D testing offers a practical way to identify eyes at near-term risk of proliferative diabetic retinopathy (PDR) beyond what HbA1c alone can reveal [11]. Interventions that flatten glucose excursions (nutrition counseling, smarter insulin titration, and new therapies like GLP-1 receptor agonists) combined with vitamin D supplementation are necessary. Early randomized data in related retinal disease indicate that vitamin D can lower VEGF and improve anatomic outcomes [28]. Because glycemic-variability instability and hypovitaminosis D are common, as international surveys suggest a percentage of 40–60% of adults with diabetes fail GV stability targets and ≥60% have sub-optimal vitamin D [6,9], even modest improvements could yield meaningful reductions in vision-threatening disease. Clinicians should therefore look beyond HbA1c: convergent evidence shows that large glucose swings [6] and profound vitamin D deficiency [29] inflict microvascular damage that mean glycaemia does not capture. In practice, we propose routine reporting of GV metrics (preferably from CGM) and annual 25(OH)D measurement, particularly in patients with early NPDR or persistently high GV [11]. These screenings can be embedded into standard diabetes visits at minimal additional cost, and earlier risk identification may ultimately reduce the need for anti-VEGF injections and vitrectomy, easing both patient burden and health-system costs. Accordingly, vitamin D status should be interpreted in conjunction with glycemic metrics; the signal is conditional on diabetes and does not substitute for glycemic control.

From a clinical and public-health implications point-of-view, vitamin-D repletion in deficient patients is a mandatory target in treatment, and causal relevance should be tested using pragmatic “treat-to-target” supplementation within standard of care, alone or combined with GV-stability interventions, where feasible, through target-trial emulation in real-world cohorts.

### Strengths and Limitations

We did not assay systemic inflammatory or oxidative-stress markers (e.g., hs-CRP, ICAM-1, 8-isoprostane), which limits mechanistic inference; residual confounding is possible despite adjustment. Future work should use ethically permissible, pragmatic treat-to-target vitamin D supplementation—alone or combined with GV-stability interventions—and target-trial emulation where feasible.

**Strengths.** This study brings a pragmatic, dual-factor perspective by assessing glycemic variability and vitamin D status in the same patients against a clinically meaningful endpoint (proliferative DR). Retinopathy was graded by masked retinal specialists using a standardized scale, and the clinical picture was complemented by SD-OCT Glycemic variability was quantified with consensus-aligned metrics (14-day CGM %CV with a ≤/>36% stability threshold) and a structured fallback, allowing comparison with existing literature. We applied broad confounder adjustment (including HbA1c, diabetes duration, BMI, smoking, triglycerides, and treatment modality), and the GV–PDR association persisted after accounting for diabetes subtype. Laboratory measurements followed established standards (25[OH]D by validated chemiluminescent assay; HbA1c by NGSP/IFCC-traceable HPLC; lipids by standardized enzymatic methods) and were obtained on the same day as the eye examination. Finally, sampling within a narrow calendar window helped limit seasonal variation in vitamin D, and the binary NPDR versus PDR classification enhances clinical interpretability.

**Limitations**. This study has several factors to consider when interpreting the findings. First, its cross-sectional design limits causal interpretation and leaves open the possibility of reverse causation (e.g., advanced retinopathy prompting treatment patterns that increase glycaemic variability), underscoring the need for prospective studies. Second, the modest sample size (particularly for vitamin D analyses) reduces power to detect interactions, so effect sizes should be viewed as approximate. Third, although analyses adjusted for subtype and treatment, some residual confounding by disease duration or severity may remain. Fourth, 25(OH)D’s seasonal variation may not be fully accounted for, although samples were collected within a two-month window to minimize bias. Fifth, this single-center Romanian cohort may reduce applicability to populations with different sunlight exposure, dietary patterns, or care pathways. Finally, OCT was descriptive, supporting plausibility but not quantifying risk. We did not assay systemic inflammatory or oxidative-stress markers (e.g., hs-CRP, ICAM-1, 8-isoprostane), which limits mechanistic inference and will be incorporated in prospective work. Also, we measured 25(OH)D once and did not capture comprehensive inflammatory/oxidative markers, so unmeasured confounding may persist; this reinforces the need for ethically permissible interventional and longitudinal designs.

Overall, these limitations suggest the associations should be viewed as preliminary signals requiring confirmation in larger, prospective studies.

**Future directions.** Prospective multicenter cohorts should confirm temporal relationships and explore dose–response effects, particularly using CGM-derived Time-in-Range and longer 25(OH)D trajectories. Future work will combine treat-to-target vitamin D supplementation (within standard of care) with systematic measurement of co-nutrient status and lifestyle proxies (dietary quality scores, outdoor-light exposure), to test whether the 25(OH)D signal persists after accounting for nutrient clustering. Interventional trials combining GV-targeted therapies with vitamin D supplementation could test whether dual optimization slows progression to PDR or reduces anti-VEGF treatment burden.

## 5. Conclusions

In adults with diabetes, both higher glycemic variability and severe vitamin D deficiency are independently associated (within adjusted models) with proliferative diabetic retinopathy, and their effects appear additive. Addressing both factors may offer a pragmatic, low-cost avenue to reduce vision loss in diabetes.

## Figures and Tables

**Figure 1 nutrients-17-03210-f001:**
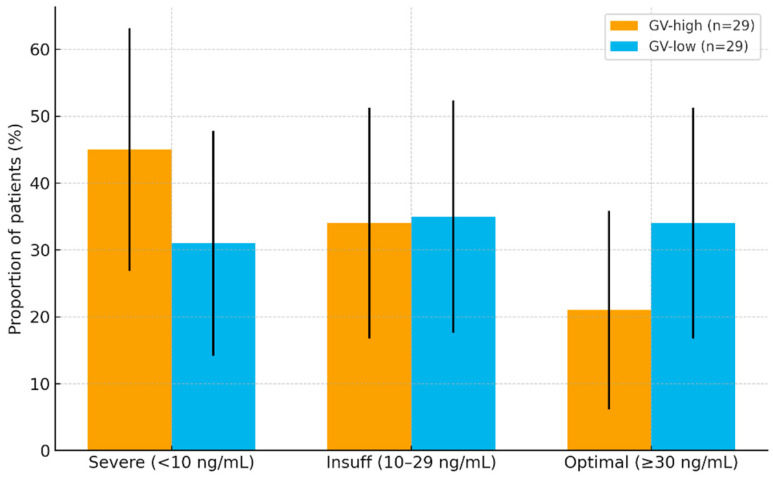
Distribution of serum 25-hydroxy-vitamin D categories in the study cohort by glycemic-variability status. Bars show proportion of patients (%) with 95% confidence intervals in GV-high (n = 29) and GV-low (n = 29) groups across severe deficiency (<10 ng/mL), insufficiency (10–29 ng/mL), and optimal (≥30 ng/mL).

**Figure 2 nutrients-17-03210-f002:**
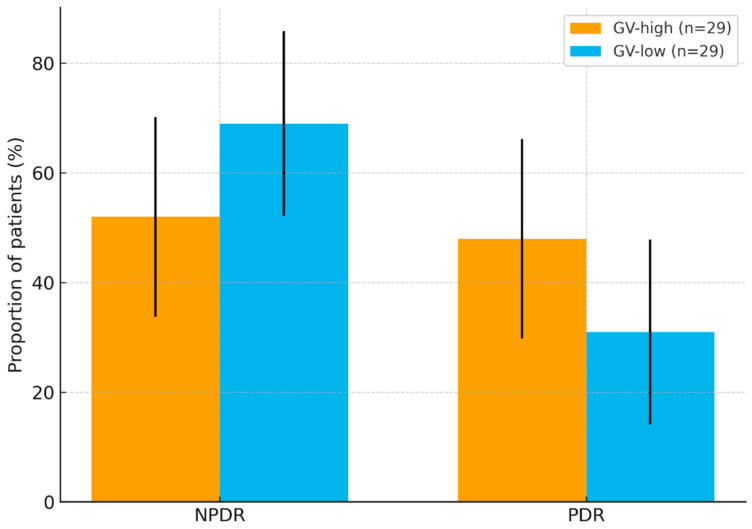
Diabetic-retinopathy stage distribution by glycemic-variability status. Bars show proportion of patients (%) with 95% confidence intervals classified as NPDR or PDR in GV-high (n = 29) and GV-low (n = 29). Unadjusted comparison for PDR: OR ≈ 2.07 (95% CI 0.71–6.06), *p* ≈ 0.18.

**Figure 3 nutrients-17-03210-f003:**
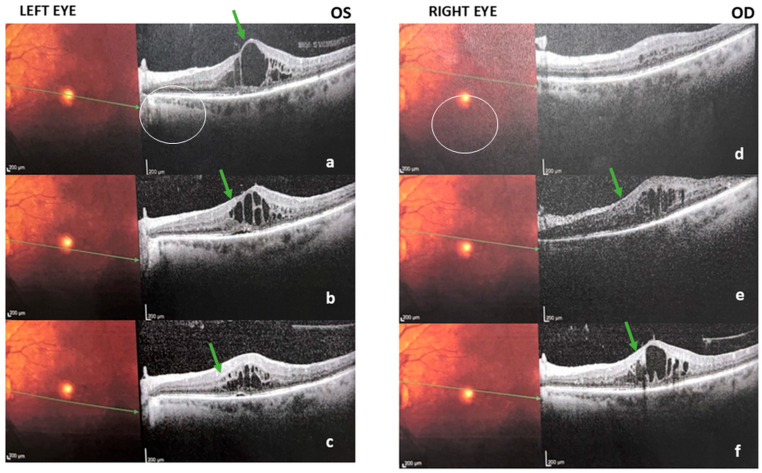
Spectral-domain OCT (three adjacent 30° horizontal B-scans) in an eye with non-proliferative diabetic retinopathy (NPDR) and diabetic macular edema (DME) from a participant exposed to high glycemic variability and severe vitamin-D deficiency. Multimodal retinal imaging—left eye (oculus sinister OS) panels (**a**–**c**) and right eye (oculus dexter OD) panels (**d**–**f**). (**a**) OS, color fundus photograph: Arrows highlight macular hard exudates and dot-blot hemorrhages. (**b**) OS, color fundus photograph: Arrows indicate clustered lipid exudates and scattered microaneurysms in the posterior pole. (**c**) OS, spectral-domain OCT (horizontal B-scan): Arrows mark intraretinal cystoid spaces within the inner nuclear layer (INL) and outer plexiform layer (OPL), consistent with diabetic macular edema (DME); the outer retinal bands and retinal pigment epithelium (RPE) remain preserved. (**d**) OD, color fundus photograph: Arrows point to posterior-pole lesions (exudates/hemorrhages) corresponding to the OCT changes. (**e**) OD, color fundus photograph: Arrows show additional macular exudation for structure–imaging correlation. (**f**) OD, spectral-domain OCT (horizontal B-scan): Arrows indicate a thin epiretinal membrane (ERM) at the inner retinal surface with no center-involved macular edema. Green arrows indicate ERM with mild traction. Circles highlights the indicated areas. Scale bar = 200 µm. Illustrative of macular involvement; DR stage (NPDR vs. PDR) was assigned by dilated fundus examination (ICDR/ETDRS), not by OCT. OCT images are illustrative and do not determine stage.

**Figure 4 nutrients-17-03210-f004:**
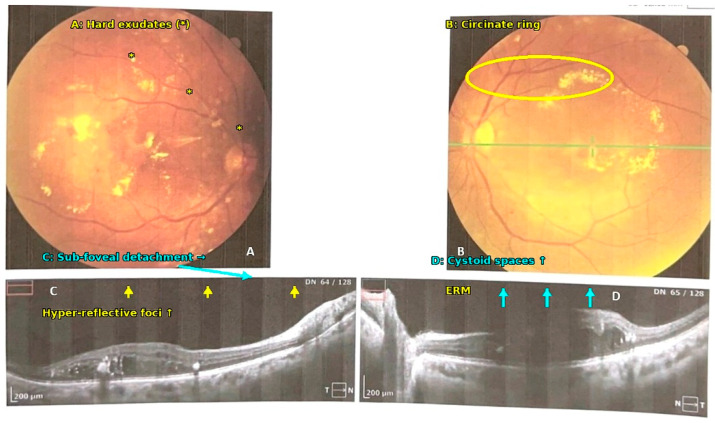
Bilateral circinate exudative maculopathy with OCT correlates in a patient exposed to high glycemic variability and severe vitamin-D deficiency. (**A**) Right-eye fundus photograph—Diffuse hard exudates (bright yellow plaques) are scattered throughout the posterior pole, with focal clustering temporal to the fovea. Several dot-blot edema is also visible, indicating active leakage. (**B**) Left-eye fundus photograph—A near-complete circinate ring of confluent lipid exudates surrounds the foveal avascular zone, a classic sign of chronic diabetic macular edema. (**C**) SD-OCT horizontal B-scan through the right fovea—Diffuse inner-retinal thickening with small hyper-reflective foci (lipid deposits) and a shallow sub-foveal neurosensory detachment. The outer-retinal bands remain continuous. (**D**) SD-OCT horizontal B-scan through the left fovea—Multiple intraretinal cystoid spaces span the inner nuclear and outer plexiform layers; scattered hyper-reflective dots correspond to the funduscopically visible hard exudates. Mild epiretinal membrane is present, exerting gentle contour distortion. Scale bars = 200 µm.

**Table 1 nutrients-17-03210-t001:** Baseline characteristics, vitamin D status, and retinopathy stage by glycemic-variability group.

Predictor.	Adjusted OR	95% CI	*p*
High GV (vs. low)	2.31	1.05–5.09	0.04
Severe vitamin D deficiency (vs. non-severe)	2.04	0.98–4.25	0.06
Both risk factors present (vs. neither)	3.88	1.35–11.1	0.012

No significant GV × vitamin D interaction was detected (*p* = 0.21), indicating additive rather than multiplicative effects. Model fit was acceptable (Hosmer–Lemeshow *p* = 0.48).

## Data Availability

The data presented in this study are available on request from the corresponding author due to their inclusion in an ongoing study.

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
