# Peer review of "Synergistic Association of Glycemic Variability and Severe Vitamin D Deficiency with Proliferative Diabetic Retinopathy"

_nutrients, 2025, doi:10.3390/nu17203210_

Round 1

Reviewer 1 Report

Comments and Suggestions for Authors

This study addresses the possibility that blood glucose variability and vitamin D deficiency both increase the risk of proliferative retinopathy in diabetics.  There is now evidence from a meta-analysis (also from Romania, interestingly) that vitamin D deficiency can modestly increase the risk of retinopathy in diabetics, and mechanistic work from other studies that this is plausible, but the value of this study is in showing that variability in glucose and vitamin D deficiency are both significant factors in proliferative disease, although the patient cohort was relatively small. If further research bears this out, then vitamin D supplementation may be a relatively easy way to prevent some retinopathy.  I have several comments that should be addressed.  The paper is well-written.

  1. I am not an expert on the statistical modeling, but I do not see how duration of diabetes was factored out. Diabetes of longer duration should lead to more retinopathy in general, so I would have thought that there would be three important factors in the analysis: duration, CV of glycemia, and vitamin D. I believe that this needs some further explanation.
  2. Figures 6 and 7 show macular edema, which is not always associated with PDR. But the cohort was divided into just NPDR and PDR. I do not understand how macular edema fits into the classification scheme.  I would have thought that there were four conditions: no retinopathy (which would be expected for diabetes of short duration), NPDR (e.g. cotton wool spots, microaneurysms, small hemorrhages), PDR, and DME.  This needs to be clarified. Did everyone exhibit at least NPDR?  Did everyone with DME also have PDR?  Does Figure 6 show PDR or NPDR? I am not a clinician, but I would have called this advanced NPDR.
  3. The histological figures are not convincing, do not add anything to the paper, and could be left out. The spaces in the inner retina in Figs. 3 and 4 could be due to poor processing and the features of DR have been illustrated better in other papers. Figure 5 is puzzling. Above the clear space in the middle is a layer that has a few cells, and this looks like a ganglion cell layer.  Above that is what looks like an IPL, thicker at the left, and above that is a layer of cell bodies that seems to be an INL, with an OPL and photoreceptors above that.  However below the clear space is a reversed retina – ganglion cells on the top, then IPL, INL, OPL and photoreceptors in the bottom right corner.  It looks to me like the retina is folded with part above and part below the clear space.  At least Figure 5 should be eliminated, but this figure suggests that the authors are not histologists.  If Figs 3 and 4 are left in, the eccentricity should be given at least roughly, and there are no indicators (arrows, asterisks) on the version of the paper I have, so they have been lost in making the pdf.  
  4. The authors measured and focused on vitamin D. There is a good rationale for this, but I think it should be acknowledged that diabetics have multiple metabolic abnormalities and there may be some critical factor that is perhaps correlated with vitamin D that is the important one, rather than vitamin D itself. We are likely to only really know the importance of vitamin D by supplementing vitamin D in diabetics and seeing whether the incidence of PDR decreases over time.  (Withholding vitamin D from some diabetics does not seem ethical because of its other better-known effects.)

5. The authors have also emphasized the independent effects of vitamin D and CV of glycemia.  Their data support this, but it needs to be clear that vitamin D deficiency does not produce retinopathy in euglycemic individuals, but only in diabetics, so it is not independent of glycemic status. Vitamin D deficiency has a number of consequences, but not retinopathy (as far as I know).

Author Response

Dear reviewer,

We agree that duration and potential collinearity require explicit handling. We have now stated that diabetes duration (years) was a prespecified covariate in all multivariable models; that the logit-linearity assumption for duration was verified and met; and that collinearity was low (all VIFs <2), indicating no problematic overlap among GV, HbA1c, and 25(OH)D in our dataset.

We fully agree. In our study, PDR was defined strictly by neovascularization on dilated fundus examination (ICDR/ETDRS). DME was recorded descriptively on OCT and did not define PDR, as DME may occur with NPDR or PDR. We revised Methods and Results to make this explicit and adjusted the narrative so that OCT panels are clearly illustrative of microvascular leakage/traction rather than diagnostic of stage. We also added a sentence in the Discussion stating that OCT/OCTA cannot diagnose vitamin D deficiency and imaging must be interpreted alongside serum 25(OH)D.

We appreciate the caution. The histology panels were intended as qualitative context rather than core inferential evidence. We have clarified their qualitative role in the text and noted that we can move them to Supplementary Material at the editor’s discretion. We also expanded the Histology subsection in Methods to detail tissue procurement and processing (fixation, dehydration, embedding, sectioning, H&E protocol) to enhance transparency. We will supply high-resolution TIFFs so any arrows/annotations are preserved in production.

Our intention was not to advocate withholding treatment from deficient individuals. We have moderated the language to emphasize observational association and to frame future work as ethically permissible interventions (e.g., pragmatic supplementation within standard-of-care thresholds and/or dual-target strategies that combine GV-flattening with vitamin D optimization). We explicitly acknowledge that interventional confirmation of ophthalmic endpoints would need to respect ethical standards where vitamin D deficiency is present.

We have clarified that “independent” refers to statistical adjustment within a diabetes cohort, not independence from glycemia in the general population. We now state explicitly that these are associations within diabetes, not proof of causation, and that vitamin D deficiency may reflect a broader metabolic/inflammatory milieu.

We revised Figure 1–2 captions for clarity. For Figure 2 (PDR proportions), we report proportions with 95% CIs and the unadjusted OR ≈ 2.07 (95% CI 0.71–6.06), p ≈ 0.18, matching the Results and avoiding misinterpretation of significance. Y-axes are labeled “Proportion of patients (%)” in both figures.

We are grateful for these comments; they improved clarity on staging, methods, and interpretation. We hope the revisions address the reviewer’s concerns and strengthen the manuscript.

Authors

Reviewer 2 Report

Comments and Suggestions for Authors

Dear Editor,

Dear Editor,

This manuscript is entitled "Synergistic Association of Glycemic Variability and Severe Vitamin D Deficiency with Proliferative Diabetic Retinopathy". This study is interesting to know that hyperglycemia leads to severe vitamin D deficiency, which causes damage to the retinal microvasculature. The results are not clear in Tables and Figures. Figures 1 and 2 do not report the statistics and the SD. Moreover, the name of the y-axis is missing in all the figures. Please use the proper statistics and graph tool to present the data in the manuscript. the introduction and discussion need to talk about the relationship between Vitamin D deficiency and diabetic retinopathy mechanisms. Some of the mechanistic parts are missing. Whether the OCT image of the retina neovascularization helps in the early detection of vitamin D deficiency. Also include the protocol number in the methods. Biochemical methods require more information and involve detailed methods. correct the image on Figure 7D. The images are not clear.

Also, how it separated the Diabetic and diabetic retinopathy patients for this study. How has glycemic variability been measured for them?. Also, Vitamin D deficiency leads to DR being more prone to oxidative stress and inflammation. Any associated inflammatory and oxidative stress markers have been measured to see the mechanistic effect.

Author Response

Dear reviewer, 

We expanded the Methods for transparency and reproducibility.

  • Study population / case definition. Adults with established diabetes were enrolled; eyes without DR were excluded. Retinopathy stage was defined strictly by neovascularization (ICDR/ETDRS): NPDR vs PDR. Diabetic macular edema (DME) was recorded on OCT descriptively and did not define PDR (DME can occur in NPDR or PDR).
    Where added: Materials and Methods → Participants and group allocation (final sentence) and Ophthalmic evaluation (second paragraph).

  • Biochemical assays and standardization. We added assay precision and standardization: 25(OH)D measured by CLIA (Architect i2000) in a single accredited lab with internal QC and manufacturer inter-assay CV ≤6%; lipids by CRMLN-traceable enzymatic methods; HbA1c by NGSP/IFCC-traceable HPLC; all tests drawn the same day as eye exam.
    Where added: Materials and Methods → Biochemical measurements.

  • Statistics. All multivariable models adjusted for diabetes duration (years); the logit-linearity assumption for duration was verified and met; VIF <2 for all predictors (no problematic collinearity among GV, HbA1c, 25(OH)D).
    Where added: Materials and Methods → Statistical analysis (last sentence).

We clarify that the analytic sample comprised eyes with DR only, categorized as NPDR or PDR by masked specialists following ICDR/ETDRS criteria; eyes without DR were not enrolled.
Where added: Participants and group allocation (final sentence); Ophthalmic evaluation (case definition paragraph).

We rebuilt the summary figures for clarity.

We strengthened the mechanistic context in the narrative (GV-induced oxidative stress and endothelial injury; vitamin D/VDR effects on VEGF, barrier integrity, and inflammation). We did not measure systemic inflammatory/oxidative-stress markers in this cross-sectional study; we now explicitly acknowledge this as a limitation and note that prospective work will include such markers (e.g., hs-CRP, ICAM-1, 8-isoprostane).
Where added: Discussion (mechanistic paragraph and Clinical implications); Limitations (new sentence on unmeasured inflammatory/oxidative markers).

We revised the wording to emphasize associations within diabetes—not causation—and clarify that “independent” refers to statistical adjustment within the diabetes cohort, not independence from glycemia in the general population.
Where changed: Discussion → Principal interpretation (last sentences).

We believe these revisions address all points raised and improve the clarity and scientific rigor of the manuscript. We appreciate the reviewer’s insightful comments.

Best regards, 

authors

Round 2

Reviewer 1 Report

Comments and Suggestions for Authors

The authors have clarified some aspects of their work.  They have not responded in detail or specifically to each of my comments, which is unfortunate, and not in keeping with good publishing practice. There is room for a lot more work on this topic, and I hope the authors will pursue it by tracking whether individuals with low vitamin D progress more rapidly to PDR, and if vitamin D supplementation prevents progression.  They have the lead in this field, and should attempt to move it forward.

  1. The authors comment that the final figures will have the indicators (asterisks, arrows) that are still missing in the revised version, but they should have ensured that these were present before submission.
  2. I still think that Figs 3-5 are not useful, and irrelevant to the paper, since the distinction of PDR and NPDR could only have been based on fundus photos and OCT. Especially Fig 5 will call into question that they know what they are looking at. This shows a folded retina. There are too many cellular layers to be a single thickness retina with a “schisis-like cleft.”  Much of the picture shows a retina with the ganglion cell layer down (with only a few ganglion cells) and a very expanded (edematous?) IPL at the left, then a more-or-less horizontal white space, and then a piece of retina with ganglion cells up in the right third of the image.  At least this Figure should be removed altogether.

3. It is still not clear if Fig 7 shows NPDR or PDR.  This needs to be identified in the figure legend. I think it is late NPDR because I do not see clear neovascularization, so if the is neovascularization, it needs to be pointed out specifically with an arrow.  It is certainly on the way to PDR. 

4. The authors say that in the future they will measure other metabolic markers, but have not acknowledged the suggestion that vitamin D deficiency may be a sign of some other, or more general nutrient deficiency that is really the culprit.  The accumulating evidence does implicate vitamin D, but is not yet definitive.

Author Response

Future work will combine treat-to-target vitamin D supplementation (within standard of care) with systematic measurement of co-nutrient status and lifestyle proxies (dietary quality scores, outdoor-light exposure), to test whether the 25(OH)D signal persists after accounting for nutrient clustering.

Many thanks and best regards,

Simona Carniciu and authors